# Effects of High-Intensity Ultrasound on the Microstructure and Mechanical Properties of 2195 Aluminum Ingots

**Yuqi Hu [1], Ripeng Jiang [1,2,*], Xiaoqian Li [1,2], Anqing Li [1] and Ziming Xie [1]**

[1]  Research Institute of Light Alloy, Central South University, Changsha 410083, China; hyqalbert@csu.edu.cn (Y.H.); meel@csu.edu.cn (X.L.); laq0823@csu.edu.cn (A.L.); xieziming@csu.edu.cn (Z.X.)
[2]  State Key Laboratory of High Performance Complex Manufacturing, Changsha 410083, China
[*]  Correspondence: jiangrp@csu.edu.cn; Tel.: +86-151-7314-9455

**Abstract:** The microstructural refinement of 2195 aluminum alloy ingots is particularly important for improving their industrial applications and mechanical properties. Combined with vacuum casting and inert gas protection, scalable high-strength ultrasonic melt processing (USMT) technology was used to manufacture 2195 aluminum alloy cylindrical ingots. Then, the influence of USMT on the main microstructural components (primary α-Al grains, secondary phase network, and precipitated particles) was studied. Our experiments show that the main microstructure of the ingot was improved after the introduction of ultrasound. Compared to the ingot formed without USMT, the size and morphology of the primary α-Al phase were optimized. The agglomeration of coarsening secondary phases can be alleviated, and the large layered secondary phase network becomes discontinuous throughout the ingot under USMT. At the same time, the mechanical properties of the solidified aluminum alloy ingots were also tested, and comparisons were made between samples formed with and without USMT. The results show that the stress concentration caused by the large area of coarse secondary phase in the ingot leads to the decrease of mechanical properties.

**Keywords:** ultrasonic treatment; 2195 aluminum alloy; microstructural refinement; mechanical properties

## 1. Introduction

By the 1990s, countries worldwide had strengthened their research on aluminum–lithium alloys, due to the implementation of the low-cost, high-efficiency, long-distance launcher program and the ultra-light fuel tank program [1]. In particular, the 2195 aluminum alloy has excellent strength, and has been used by the United States to replace 2219 alloy to form the outer storage box of a space shuttle, thus increasing its aerospace carrying capacity by 3.4 t; furthermore, the liquid hydrogen and liquid oxygen storage tanks on the rocket are made of 2195 aluminum–lithium alloy [2]. In order to be further processed into aerospace parts, aluminum alloys are usually cast into suitable shapes (high-quality ingots are typically required), and the final products are made after forging and rolling. Therefore, the casting process is one of the key processes affecting the final structure and properties of the 2195 aluminum alloy ingot. Recently, in order to improve the quality of ingots, researchers have applied many related technologies in the casting process, including physical methods (e.g., mechanical or electromagnetic stirring) and chemical methods (e.g., the addition of nucleating agents). Among these methods, ultrasonic treatment (UT) has been proven to have great potential, due to its high efficiency in refining the solidified structure and changing mechanical properties [3–5].

AC7A aluminum alloy adopts USMT to process the melt, and the structural properties, corrosion resistance, and mechanical properties of the alloy before and after the treatment have been explored. The results showed that the solidification process under the action of ultrasound can lead to fine spherical grains and metal compounds [6–9]. Eskin et al. [10–14] found that the application of ultrasonic treatment to the aluminum melt can effectively

refine the solidification structure. Part of the gas impurities in the melt was removed, and a crack-free 2324 aluminum alloy ingot with a diameter of 1100 mm was successfully prepared. Huang [15], in the process of a high-purity aluminum solidification experiment in a crucible, through the application of ultrasonic treatment and modeling analysis, found that the heterogeneous nucleation induced during the ultrasonic treatment can produce the same fine crystal effect as spontaneous nucleation and that the effect of tissue refinement decreases as the distance from the end face of the ultrasonic radiation rod increases. Li [16] conducted a simulation analysis on the distribution of the ultrasonic cavitation area and found that the ultrasonic cavitation area will increase in a sheet shape with an increase in the insertion depth of the ultrasonic rod, while the strength will decrease, and successfully prepared a variety of 2xxx and 7xxx size ultrasonic large-diameter aluminum alloy ingots. The effects of ultrasonic power, frequency, ultrasonic rod insertion depth, and other parameters on the quality of large-diameter ingots have also been explored, and the best ultrasonic experimental parameters to realize industrial production was proposed [17]. Generally, the main mechanisms proposed in the literature to explain the role of UT in alloy grain refinement can be divided into cavitation [18–20] and acoustic flow [21].

Due to the active characteristics of lithium, aluminum–lithium alloy ingots are prone to loss during the preparation process [22–24]. At this time, the advantages of the ultrasonic casting process in melt stirring and degassing become evident. This study provides insights into the influence of the ultrasonic power of high-intensity ultrasonic radiation on the microstructure and mechanical properties of large ingots.

## 2. Experiment

### 2.1. Experimental Equipment and Scheme

The schematic diagram of the ultrasonic system, the deployment of the ultrasonic probe, the typical industrial-grade vacuum casting site, and the combination of the ultrasonic casting site are shown in Figure 1. The ultrasonic system included an ultrasonic generator with a maximum output power of 2 kW, a 20 kHz piezoelectric transducer with an air cooler, an ultrasonic amplitude converter, and an ultrasonic vibrator composed of a titanium alloy sonotrode with a tip diameter of $\varphi50$ mm (the dimensions of the titanium alloy sonotrode are $\varphi50 \times 7250$ mm). The chemical composition of the titanium alloy sonotrode can be obtained from [25]. The maximum vibration amplitude measured on the end face of the ultrasonic generator was 20 µm. Before being immersed in the alloy melt, the ultrasonic probe was pre-heated at 300 °C, in order to prevent the impression of chilling on the system, and the power supply was adjusted to achieve a stable state.

The melting and casting process is described as follows:

Before casting, the air was pumped out from the melting furnace cavity using a vacuum pump unit, where a resistance vacuum gauge was used to measure the gas pressure in the furnace cavity, to 1 pa (standard atmospheric pressure = $10.1 \times 10^4$ pa). Argon gas was then pumped in to bring the chamber to standard atmospheric pressure and control the oxygen in the furnace chamber, as much as possible. The raw alloy materials were melted in a resistance furnace under an argon protective atmosphere. Then, the manual slag removal, mechanical stirring, and nozzle inert gas flotation (NIF) methods were applied to refine the melt. The size of the melt in the graphite crucible was a cylindrical shape with a diameter of 3500 mm and a depth of 40 mm. Then, the ultrasonic assisted casting system was applied to the treated aluminum melt. In the ultrasonic casting process, the ultrasonic probe used was immersed 20 mm below the surface of the melt, based on previous research [25]. After 30 min of ultrasonic treatment, the melt temperature was set to 725 °C, the output power was set to $1000 \pm 50$ W [26] in the furnace cavity, and the aluminum melt was introduced into the mold for casting.

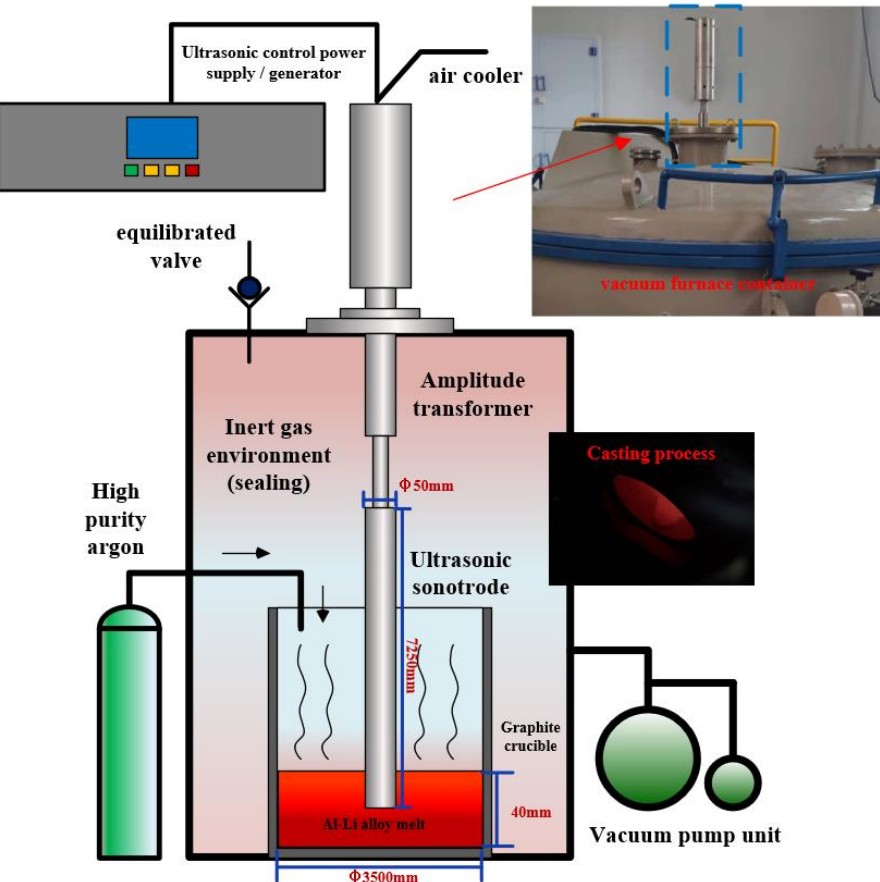

**Figure 1.** Schematic diagram of ultrasonic system and vacuum system.

## 2.2. Experiment Characterization Methods

After natural aging, a circular plate with a thickness of 15 mm was cut transversely from three parts of the two ingots. As shown in Figure 2, a sample for microstructure analysis was obtained at a specific position in the radial direction. In order to test the mechanical properties, samples were taken for the radial and axial positions of the ingot at the same time.

All samples were ground and polished according to standard metallographic techniques. The microstructure was characterized by a scanning electron microscope (SEM; TESCAN, MIRA3 LMH/LMU, Carl Zelss, Shanghai, China) equipped with an energy dispersive spectrometer (EDS). which was used to analyze the composition of each phase in the sample. In order to meet the requirements of metallographic phase inspection, all samples were anodized for 50 s using a stainless steel cathode at 20 VDC using Barker reagents (200 mL 32% $HBF_4$ in 800 mL distilled water). Metallographic phase observation was performed under polarized light with a ZEISS optical microscope equipped with an automatic Zeiss AxioVision image analyzer (Carl Zelss, Shanghai, China). The grain size was analyzed based on the linear intercept method (ASTM 112-10). The precipitating particles were examined by a scanning electron microscope (SEM TESCAN, MIRA3 LMH/LMU) equipped with an energy dispersion spectrometer (EDS). ICP-AES (Inductively Coupled Plasma Atomic Emission Spectrometry) (Spectro Analytical Instruments GmbH, Kleve, Germany) was used to analyze the composition of a specific position of the ingot, in order to obtain the composition distribution of the entire ingot. The distribution of secondary networks and precipitating particles was quantitatively analyzed using image analysis software (Image Pro Plus, IPP, Media cybernetics, Rockville, MD, USA). Figure 2 shows the tensile test sample (GB/T 16865-2013) obtained at a representative location, such as

schematics and dimensions, and the sample is tested using the Instron 3369 mechanical tester at a minimum load rate of 1.5 mm/min.

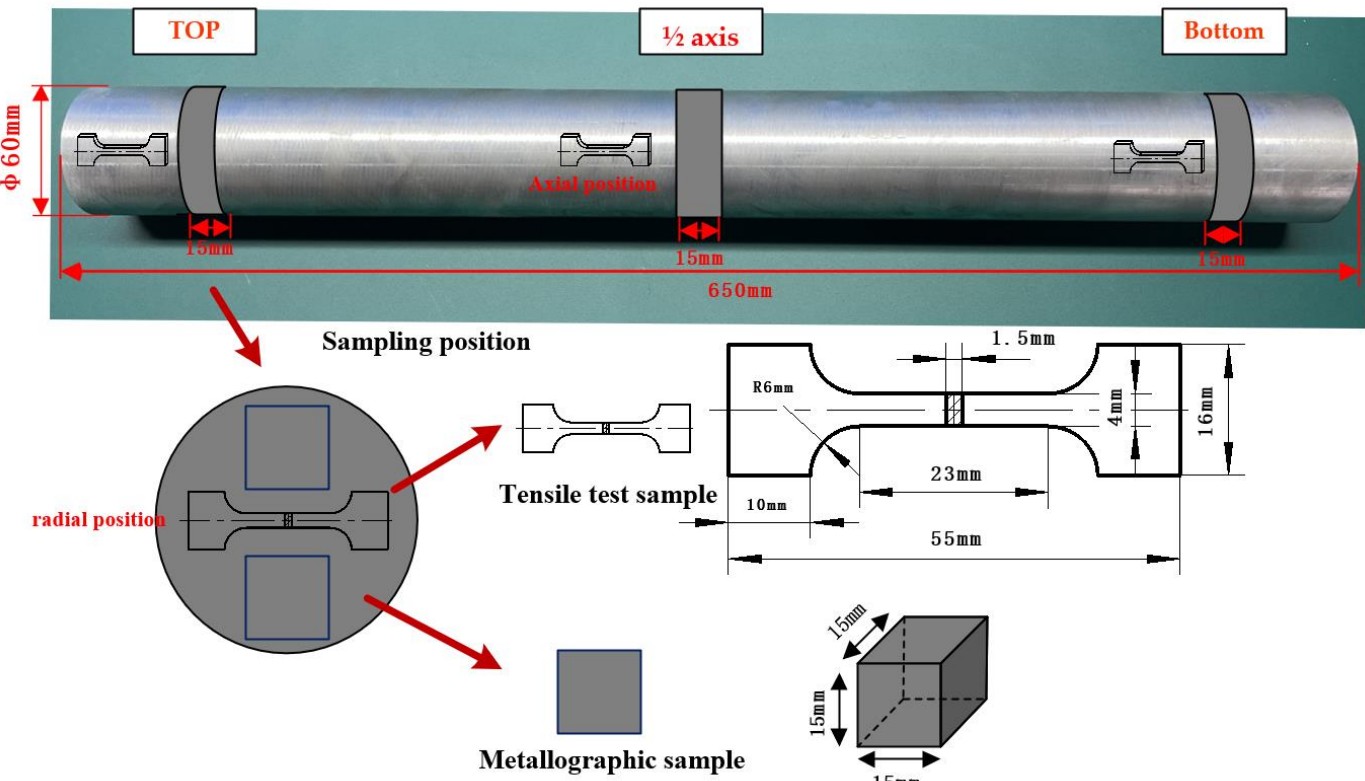

**Figure 2.** Schematic diagram showing representative positions of microstructure samples and tensile test samples taken from the ingot.

### 2.3. Materials

The dimensions of the AA2195 aluminum–lithium alloy ingot prepared in the experiment (after peeling) was as follows: a cylindrical ingot with a diameter of φ60 mm and a length of 600 mm. The average composition content of the ingot was detected by ICP-AES (Inductively Coupled Plasma Atomic Emission Spectrometry), as detailed in Table 1. Considering the ingot with ultrasonic treatment and the ingot without ultrasonic treatment, the elemental content of the two ingots was similar. Therefore, the influence of the elemental content of the test material on the analysis results can be reduced.

**Table 1.** Components of 2195 aluminum–lithium alloy (mass fraction/%) [22].

|  | Li | Cu | Mg | Ag | Zr | Mn | Ti | Al |
|---|---|---|---|---|---|---|---|---|
| AS with USMT | 1.038 | 4.083 | 0.410 | 0.392 | 0.131 | 0.241 | 0.0012 | - |
| AS without USMT | 1.051 | 4.076 | 0.397 | 0.404 | 0.128 | 0.221 | 0.0008 | - |
| AA2195 [22] | 0.8–1.2 | 3.7–4.3 | 0.25–0.8 | 0.25–0.6 | 0.08–0.16 | <0.25 | - | Bal. |

## 3. Results and Discussion

### 3.1. Microstructure of α-Al Phase

In Figure 3, the metallographic photos of three specific positions in the ingot are shown. The three positions are the top of the ingot, the middle position of the ingot, and the bottom of the ingot. It can be seen from the figure that there were significant differences in the α-Al grain size between ingots from different processes. Indeed, it can be clearly distinguished that the grain size of the α-Al phase in all three positions of the ingot formed

using USMT technology was much smaller than that of the α-Al phase in the ingot formed without using USMT technology.

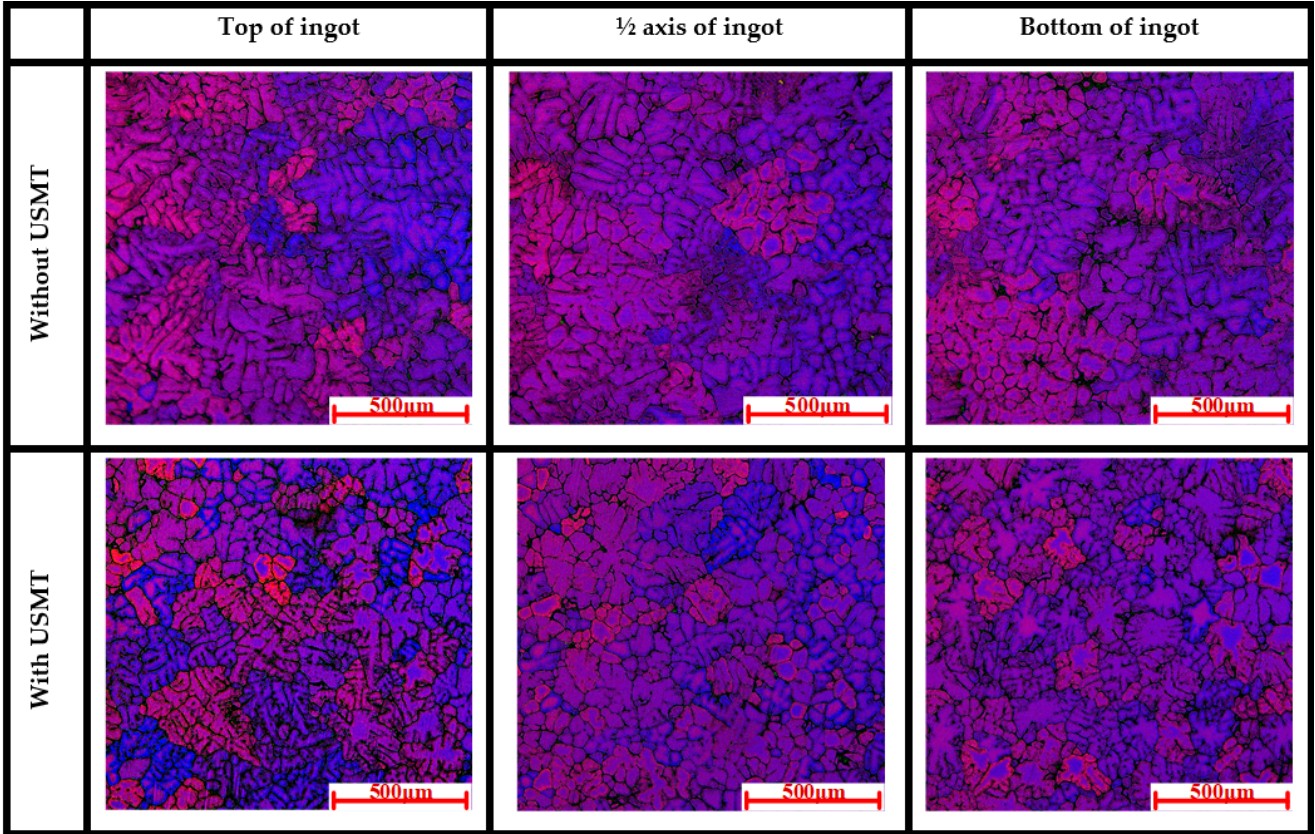

**Figure 3.** Optical micrographs at specific sampling positions in ingots formed with and without USMT.

The linear intercept method (ASTM 112-10) was applied to quantify the average size of the α-Al phase at three locations in two ingots, Figure 4. Normally, in a cast ingot, the average size of α-Al grains decreases continuously from the middle part of the ingot to the two ends (top and bottom); this feature can be clearly seen in Figure 3. However, the rate of change in the α-Al grain size for the ingot formed using the ultrasonic melt treatment (USMT) technique was smaller than that of an ingot formed without use of the ultrasonic melt treatment (USMT) technique; in particular, the rate of change was reduced by 58%. For example, the average grain size of the α-Al grains on the top of the ingot formed with ultrasonic melt treatment (USMT) was 221.08 μm, while the α-Al grain size of the ingot without ultrasonic melt treatment (USMT) was 362.79 μm. In the half axis position, the α-Al grain size of the two ingots increased, where the values were 245.86 (USMT) and 407.75 μm (without USMT). In contrast, the α-Al grain sizes at the bottom of the ingot were reduced to 196.04 (USMT) and 332.57 μm (without USMT). Table 1 lists the refinement efficiency of USMT α-Al grains in representative positions. The refinement efficiency was more than 36% in all positions and the highest refining efficiency was in the middle of the ingot (at 41.1%).

The main improvement mechanism of USMT can be summarized as cavitation-enhanced heterogeneous nucleation, since heterogeneous nucleation particles and molten elements are homogenized under the action of ultrasound [27–30]. During the cooling process of the ingot, the temperature varies unevenly from the center to the edge, which affects the microstructure of the ingot. At the edge of the ingot, the heat can be quickly taken away by the external medium, such that the α-Al grains will crystallize immediately [31]. Larger undercooling can increase the nucleation rate of α-Al grains [32]. Therefore, fine equiaxed grains are produced at the edge of the ingot; however, there was a difference

in the average grain size of α-Al in ingots with and without USMT. We observed that the cavitation effect and acoustic current effect produced by ultrasound have an intuitive influence on the refinement of the α-Al grains in the entire ingot, as shown in Table 2.

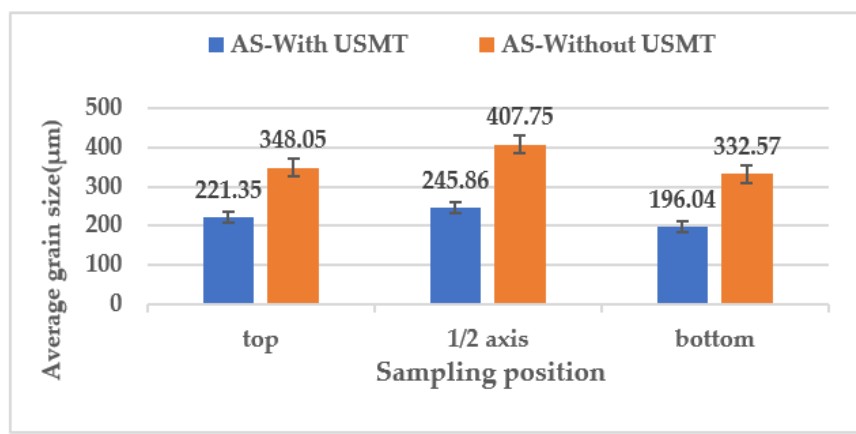

**Figure 4.** The grain sizes of the ingots with and without USMT at different sampling positions.

**Table 2.** Average size (AS) and refining efficiency (RE) of the α-Al grain.

|  | Top | Half Axis | Bottom |
| --- | --- | --- | --- |
| AS with USMT(μm) | 221.35 | 245.86 | 196.04 |
| AS without USMT (μm) | 348.05 | 407.75 | 332.57 |
| RE (%) | 36.4 | 39.7 | 41.1 |

Although USMT does not act directly on the solidification process of aluminum melts, under the action of ultrasonication during the smelting process, aluminum melts produce a large number of empty bubbles. The bursting process of the cavitation bubble will release a huge amount of energy, and then generate a powerful micro-jet impact. Continuous micro-jet streams remove impurities from the surface of the particles, thereby increasing the wetness of the particles used as hetero-shaped nucleation points and increasing the nucleation rate [33,34]. These nucleated particles may come from naturally active particles present in the melt, or from the original formation. There is already sufficient literature which has proven that TiAl$_3$, ZrAl$_3$, NbAl$_3$, and/or BTi$_2$ are valid nucleation points. As the nucleation rate of the melt is further improved, the grain refinement of the α-Al grain is enhanced [35,36].

Due to the introduction of ultrasonic waves before the solidification of the ingot, a large number of cavitation bubbles collapse and rupture during this process, resulting in relatively severe cavitation effects. At the moment when the cavitation bubble bursts, the released energy locally generates a high-temperature and high-pressure pulse. According to the Tzanakis and Khavari [37,38], the shock wave speed can reach $1464 \pm 5 \text{ ms}^{-1}$. Along with the formation of the pulse wave, shear stress appears in the liquid around the bubble, which has a strong mechanical effect [39]. It has a stirring effect on various elements in the melt and heterogeneous nucleating particles, thus improving their distribution in the melt and improving the ingot quality.

### 3.2. Distribution Characteristic of Secondary Phase

The main secondary phase of 2195 aluminum alloy is mainly α-Al + T1-Al2CuLi + θ-Al2Cu [5]. Figure 5 shows the morphology and distribution of the α-Al + T1-Al2CuLi + θ-Al2Cu secondary phase in the two ingots. It can be seen that the mixture of T1-Al2CuLi and θ-Al2Cu gather together along the grain boundaries of the α-Al grains, and the small spherical θ-Al2Cu precipitated particles are scattered in the α-Al grains of the two ingots. The acicular T1-Al2CuLi precipitated particles with sub-micron scale are dispersed inside

the α-Al crystal grains, such they could not be displayed in the SEM fiber photo. In Figure 5, it can be clearly observed that the size of the secondary phase in the ingot applied with ultrasonic waves was significantly smaller than the size of the secondary phase in the conventional ingot in the samples at all positions.

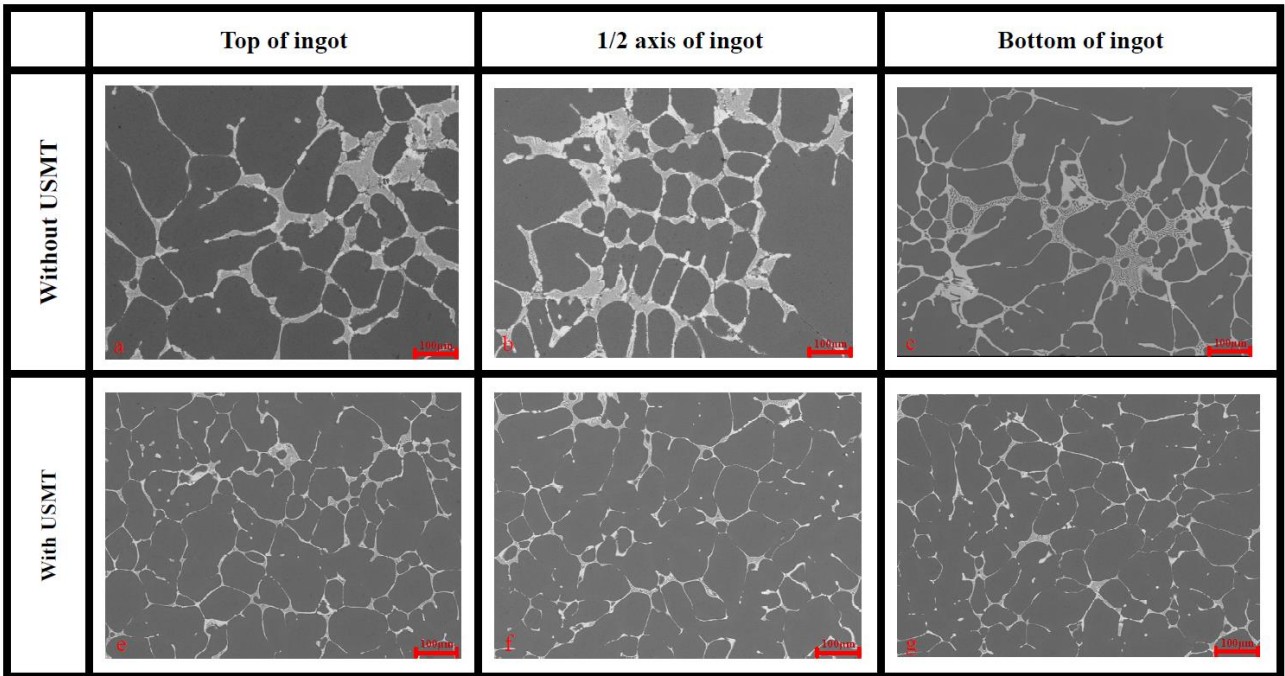

**Figure 5.** Morphology and distribution of secondary phases and precipitated particles at different sampling positions in ingots formed with and without USMT.

Quantitative descriptions of the changes in the distribution and location of secondary phases and precipitated particles, by the Image-Pro Plus (IPP) software, are shown in Figure 6, including the area fraction of the coarsened secondary phases along the grain boundary area (>20 μm$^2$) and the precipitation of the grain inner area (<20 μm$^2$) density of particles (number of particles per unit area; μm$^2$).

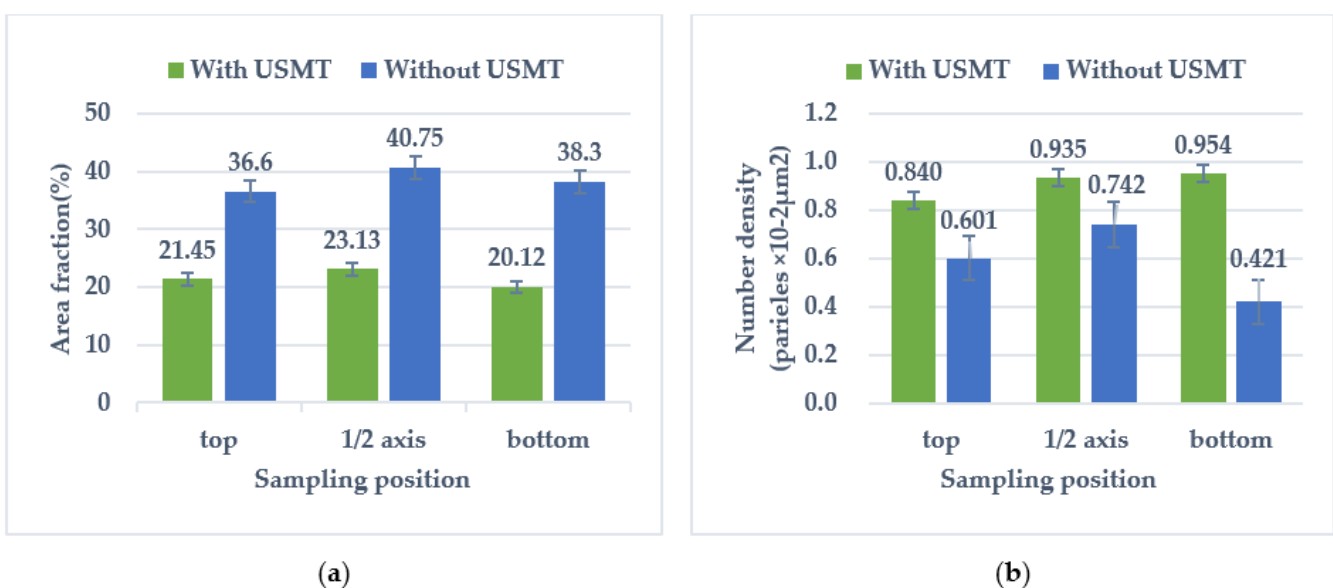

**Figure 6.** (**a**) Area fraction of the coarsening eutectic phase; and (**b**) the density of the precipitated particles in ingots with and without USMT.

The statistical results after quantification are shown in Figure 6. As shown in Figure 6a, the area fraction of the coarsening secondary phase in the ultrasonicated ingot changed little overall: the overall level was maintained at around 20% and the maximum appeared at the position of the half axis, with a value of 23.12%. However, the area fraction of the coarse precipitate phase of the sample formed without USMT was significantly higher than that of the ultrasonicated ingot, and the peak value also appeared at the half axis position (with a value of 40.75%). For conventional and ultrasonicated ingots, the overall trend of the area fraction of the coarsening secondary phase network gradually decreased from the half axis position to the two ends (i.e., top and bottom) of the ingot, and the two ends gradually decreased to the lowest value (20.12%); however, the rate of change in the same ingot was within an acceptable range.

As shown in Figure 6b, the density of precipitated particles in the ultrasonicated ingots changed slightly: the minimum value at the top was 0.840%, and it gradually increased to 0.954% at the bottom. For the corresponding value in the ingot without USMT, the maximum value at the half axis position was 0.742%, while the minimum value at the bottom position was 0.421%.

Table 3 lists the reduction rate (η) of the area fraction (AF) of the coarsening precipitate phase at representative positions. The area fraction of the secondary phases of ultrasonicated ingots was generally lower than that of conventional ingots, and samples were taken at representative locations; in particular, the conversion rate at the bottom position was reduced by 47.5%, compared to conventional ingots formed without USMT. The overall reduction rate remained above 41%.

**Table 3.** Reduction rate (η) of the area fraction (AF) of the coarsening precipitate phase.

|  | Top | Half Axis | Bottom |
| --- | --- | --- | --- |
| AS with USMT (μm) | 21.45 | 23.13 | 20.12 |
| AS without USMT (μm) | 36.60 | 40.75 | 38.30 |
| η (%) | 41.4 | 43.2 | 47.5 |

The EBSD phase mapping in Figure 7 shows the distribution of the sub-micron T1 phase in the crystal grains of the top sample: Figure 7a is from the USMT ingot and Figure 7b is the ingot formed without USMT. Obviously, even in the samples formed without USMT, a large number of T1 clusters gathered around the grain boundaries, while in the samples with USMT, the T1 phase was dispersed. The content of the T1 phase of the USMT ingot was 3.4%, as measured by mapping, while the content of the T1 phase of the ingot without USMT was 3.24%.

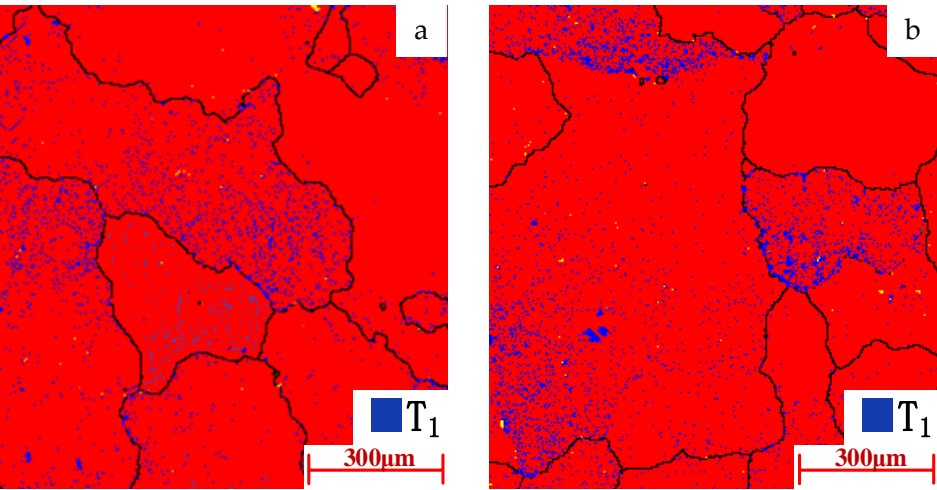

**Figure 7.** EBSD phase mapping showing the distribution of the secondary network at the top of the ingot formed (**a**) with USMT and (**b**) without USMT.

As the main coarsening secondary phase, the Cu and Li elements are related. Therefore, it is very important to control the distribution of Cu and Li in the ingot. The index $\triangle C$ is defined to evaluate the macro-segregation degree of Cu and Li [38], as determined by Equation (1)

$$\Delta C = \frac{C - C_0}{C_0} \tag{1}$$

where $C$ represents the Cu and Li content at a certain test location, the element content results come from the ICP-AES (Inductively Coupled Plasma Atomic Emission Spectrometry) test, and $C_0$ is the average content of these elements in the ingot. According to the definition of $\triangle C$, the macro-segregation of Cu and Li in the ingot axial direction was determined, as listed in Table 4.

**Table 4.** Macro-segregation ($\Delta C$) of Cu and Li at representative positions.

|     |                     | Top     | Half Axis | Bottom  |
| --- | ------------------- | ------- | --------- | ------- |
| Cu  | AS with USMT (μm)   | −0.035  | 0.030     | 0.025   |
|     | AS without USMT (μm) | −0.057  | 0.075     | −0.015  |
| Li  | AS with USMT (μm)   | −0.025  | 0.020     | 0.110   |
|     | AS without USMT (μm) | −0.315  | 0.260     | 0.150   |

In conventional ingots and ultrasonicated ingots, the Cu and Li content at the half axis was higher than at other positions, leading to positive macro-segregation of Cu and Li at this position. Comparing the segregation degree of each element at the half axis position, that of the ultrasonic ingot was lower than the conventional ingot. As for the lower negative macro-segregation at the top and bottom of the ingot, under USMT, the macro-segregation of Cu and Li was alleviated at all sampling positions.

The refinement of the secondary phase can be attributed to the following mechanisms: (a) the forced stirring and homogenization of the acoustic flow under the cavitation effect will distribute the Cu and Li atoms more evenly throughout the ingot during the solidification process [40], reducing the formation of the coarse secondary phase; and (b) the ingot formed under the application of USMT, the decrease in the area fraction of the secondary phase network can be attributed to the refinement of $\alpha$-Al grains. During the solidification of the ingot, the refined $\alpha$-Al phase provides more grain boundaries and reduces the interfacial energy. The secondary phase can be precipitated in more locations, and the continuity is reduced [41]. In addition, under USMT, more dendritic $\alpha$-Al grains are transformed into equiaxed grains. The dendritic bridging ability is reduced [42] which, together with the evenly distributed solute, inhibits the formation of a secondary phase network.

### 3.3. Mechanical Properties

We sampled according to the representative positions marked in Figure 2, and obtained axial and radial samples at the top, half axis, and bottom of the ingot, in order to test the mechanical properties of the ingot. Figure 8a–f shows the ultimate tensile strength (UTS), yield strength (YS), and elongation of the ingot at different positions along the radial and axial directions, respectively. The reason for this is that there is a risk of anisotropy in this series of alloys [43].

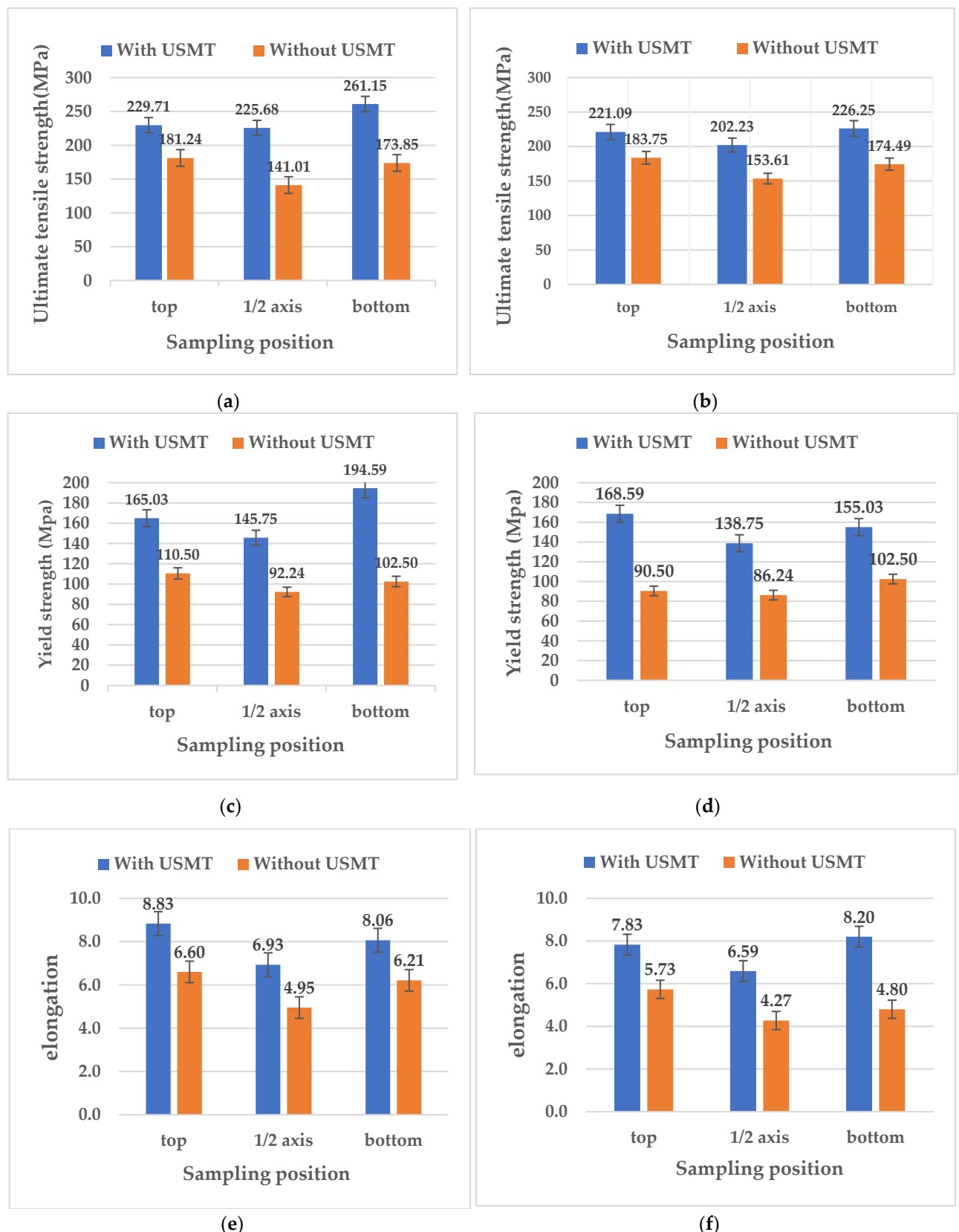

**Figure 8.** (**a**,**b**) Ultimate tensile strength (a, axial; b, radial) processed under different ultrasonic; (**c**,**d**) yield strength (c, axial; d, radial); and (**e**,**f**) elongation, determined at different positions of the ingot (e, axial; f, radial).

The UTS, YS, and elongation increased continuously from the half axis position of the ingot to the two ends of the ingot formed with USMT. When comparing the axial

and radial directions at the same time, in terms of the UTS, YS, and elongation, the axial direction performances of these three indicators were slightly higher than those in the radial direction; however, the highest values of UTS, YS, and elongation for the samples formed without USMT appeared from the top of the ingot to the position of the half axis, then decreased to the minimum, and gradually increased from the position of half axis to the bottom of the ingot. Similarly, when comparing the performance difference between the radial and axial directions, the axial direction had better mechanical properties. Thus, USMT improved the UTS, YS, and elongation of the ingot, to a certain extent. At both ends of the ingot, the UTS increased from 181.24 (without USMT) to 261 MPa (with USMT), and the YS increased from 110.5 (without USMT) to 194.59MPa (with USMT) at both ends of the ingot. The elongation rate also increased significantly, from 4.8 (without USMT) to 8.2% (with USMT), an increase of 70.8%.

The increase in tensile strength can be attributed to the increase in the uniformity of the microstructure, in which the composition phases in the ingot to which USMT is applied, appear to be fine and uniformly distributed.

We adopted the Hall–Petch formula presented in Equation (2)

$$\sigma_s = \sigma_0 + Kd^{-\frac{1}{2}} \tag{2}$$

where $\sigma_s$ is the yield strength, $\sigma_0$ is the material constant, $K$ is the Hall–Petch slope, and $d$ is the average grain size [43]. The strength of the material is inversely proportional to the grain size [44]. Therefore, with a finer grain of the ingot, better mechanical properties can be obtained [45]. furthermore. According to Reference [46], ultrasound can promote the diffusion of solute elements in the matrix, increasing the solid solubility of the matrix to the solute elements, which also explains the observed microstructure of UST ingots, and the reduction of coarse secondary phases. The increase in ingot yield strength using UST is likely due to an increase in solute, which leads to solid-soluble reinforcement in the casting state.

Figure 9 shows the fracture morphology of the tensile samples in the axial and radial portions at the bottom positions of the two ingots. As shown in Figure 9c,d, the cleavage surface and rough tear ridge, representing typical brittle fracture characteristics, can be seen on the fracture surface of the ultrasonically processed sample. The fracture surface of this sample was dominated by transgranular fracture, and the area fraction of the coarse tear ridge exceeded 40%. Compared with the cross-section of the ultrasonic ingot, the transgranular fracture was greatly reduced, and the fracture mode was mainly intergranular fracture (smooth surface); see Figure 9a,b. In the radial section, the area of cleavage and the transgranular fracture surface were reduced, but still accounted for most of the fracture surface; see Figure 9b. However, in the axial section, the intergranular fracture surface (smooth tearing edge) and small and deep pits were evenly distributed on the fracture surface, indicating a mixed fracture mode of toughness and cleavage fracture; see Figure 9a.

However, the improvement of the mechanical properties of the 2195 ingot cannot be simply attributed to the reduction of size of the primary α-Al grains. The average size of α-Al grains in ultrasonic ingots was smaller than that in conventional ingots, and the strength and elongation were improved, to a certain extent. Compared with conventional ingots, the density of particles in the grains of ultrasonic ingots was higher, and the area fraction of the coarse secondary phase was smaller. According to the reference, the agglomeration of the coarsening second phase is considered to be the first choice for crack initiation during the deformation process, which is prone to stress concentration [47]. The second phase, which reduces the coarsening, plays an important role in improving the mechanical properties. Normally, the increase in strength comes at the expense of ductility; however, the fine α-Al grains and uniformly distributed second phase can increase both the strength and ductility [48]. In addition, finer spherical particles can reduce the accumulation of damage during deformation, thereby improving the ductility of aluminum alloys [49]. According to the tensile test data, the samples with ultrasonically processed melt had higher yield

strength values. The reason for this may be grain boundary strengthening (Hall–Petch effect) and precipitation strengthening. [50]. Therefore, under USMT, as the coarsening secondary phases decreases and the refined secondary phases increases, the elongation of the sample increases accordingly.

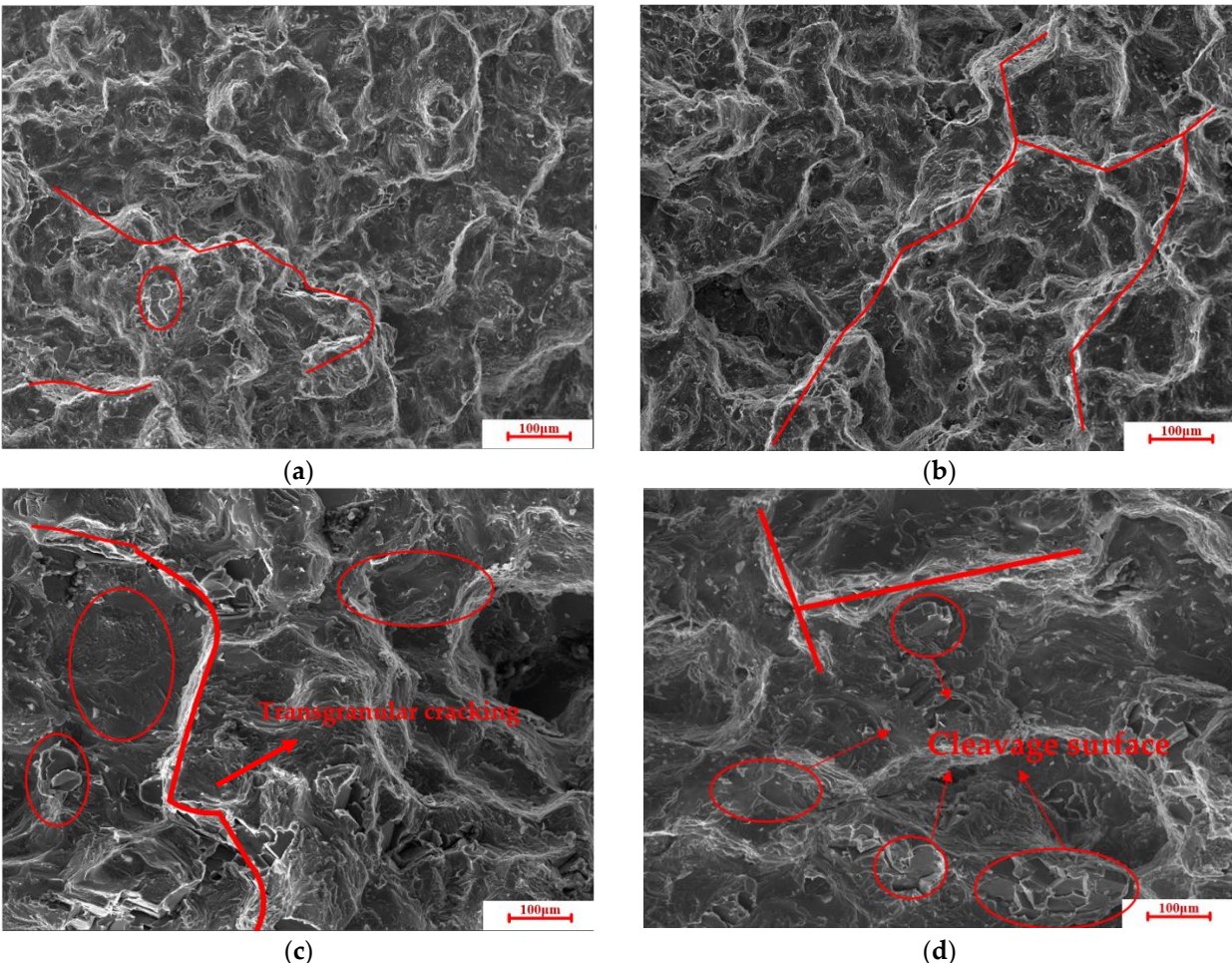

**Figure 9.** SEM image of the tensile fracture surface of the ingot using USMT: (**a**) axial part; (**b**) radial part; (**c**) axial part of the ingot without USMT; and (**d**) radial part without USMT.

## 4. Conclusions

The effect of scalable USMT on the structure and mechanical properties of 2195 aluminum–lithium alloy ingots was systematically studied.

The current main findings can be summarized as follows:

1. Compared with conventional ingots, the size of primary α-Al grains was refined under ultrasonic treatment. As the coarse α-Al grains are transformed into fine dendrites and a few equiaxed grains, the maximum refining efficiency of the primary α-Al grains reached 41.1%.

2. Under the action of strong convection generated by the ultrasound current, the solid solution copper and lithium elements were evenly distributed in each part of the whole ingot. This was the main reason for reducing the coarsening secondary phase network and increasing the density of fine θ-Al$_2$Cu and T1 particles. In addition, the refined α-Al grains reduce the degree of bridging, while the uniform distribution of solutes also limits the formation of the coarse secondary phase network. At the half axial position, the area fraction reduction rate of the coarsening secondary phase reached 43.2%.

3. The coarsening of the secondary phase network has a great influence on the mechanical properties of 2195 aluminum alloy ingots. Compared with conventional ingots, the area fraction of the secondary phase was higher, due to its coarsening. With the addition of ultrasound to the ingot, the coarse secondary phase network of the whole ingot was reduced and the mechanical properties were correspondingly improved, and the elongation was significantly increased. This shows that reducing the brittleness and coarsening of the secondary phase can improve the ductility of large ingots. The results also showed that USMT has a certain degassing effect on the melt, which has an influence on the mechanical properties of the ingot; further research is needed in future research to assess this effect.

**Author Contributions:** Conceptualization, R.J. and X.L.; Methodology, Y.H. and R.J.; software, Y.H. and Z.X.; Validation, R.J., Y.H. and A.L.; Formal analysis, R.J. and Y.H.; Investigation, Y.H.; Resources, R.J. and X.L.; Data curation, R.J. and Y.H.; Writing—original draft preparation, Y.H.; Writing—review and editing, R.J.; Visualization, Y.H. and A.L.; Supervision, R.J.; Project administration, R.J.; Funding acquisition, R.J. and X.L. All authors have read and agreed to the published version of the manuscript.

**Funding:** This research was funded by the National Natural Science Foundation of China, grant no. 51805549.

**Data Availability Statement:** Not Applicable.

**Conflicts of Interest:** The authors declare no conflict of interest.

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
