# Peer review of "Effects of High-Intensity Ultrasound on the Microstructure and Mechanical Properties of 2195 Aluminum Ingots"

_metals, doi:10.3390/met11071050_

Round 1

Reviewer 1 Report

Please find a reviwer's comment as attached.

Author Response

Dear editors and reviewers,

Thank you a lot for your helpful comments and constructive suggestions on our manuscript. We have learned the comments and suggestions earnestly and made careful corrections which we hope to gain your approval.The responses to the reviewer’s comments and the main corrections are listed point by point as follows,please see the attachment.

Reviewer 2 Report

An interesting manuscript is suitable for this journal. Many important references are missing. Better quality and higher image resolution is required. Numerous careless grammatical mistakes! English needs significant improvement. Interpretations and relevant discussion is not fully justified by the presented data. Thus, a major revision is required before this paper could be suggested for publication.

“Among these methods, ultrasonic treatment (UT) has proven to have great potential because of its high efficiency in refining the solidified structure and changing mechanical properties [3.4].” please referred to this paper: D. Eskin et al. “Fundamental studies of ultrasonic melt processing” Ultrasonics Sonochemistry 52 (2019) 455-467

“Related research on the effect of high-intensity ultrasonic treatment on the structure, corrosion and mechanical behavior of AC7A aluminum alloy” improve this sentence

“The results show that the solidification process under the action of ultrasound can obtain fine spherical grains and metal compounds [5]” please also add: 1) T. Subroto et al. “Numerical modelling and experimental validation of the effect of ultrasonic melt treatment in a directchill cast AA6008 alloy billet” Journal of Materials research and technology 12 (2021) 1582-1596; 2) GSB Lebon et al. “Numerical modelling of acoustic streaming during the ultrasonic melt treatment of direct-chill (DC) casting” Ultrasonics sonochemistry 54 (2021) 171-182; 3) T. Subroto et al. “Structure refinement upon ultrasonic melt treatment in a DC casting launder” JOM 72 (2020) 4071-4081

“Eskin et al. [6-8] found that the application of ultrasonic treatment in the aluminum melt can effectively refine the solidification structure” please add the following paper: doi:10.1007/978-3-319-94842-3_5

“and proposed the best ultrasonic experimental parameters to realize industrial production.” Please also add this reference that explains the important parameters during ultrasonic treatment:  doi: 10.1016/j.matdes.2015.11.010

“Generally, the main mechanisms proposed in the literature to explain the role of UT in alloy grain refinement can be divided into cavitation and acoustic flow.”

For cavitation please refer to: doi:10.1016/j.matprotec.2015.10.009; GSB Lebon “Numerical modelling of ultrasonic waves in a bubbly Newtonian liquid using a high-order acoustic cavitation model” Ultrasonics Sonochemistry 37 (2017) 660-668; GSB Lebon “Experimental and numerical investigation of acoustic pressures in different liquids” Ultrasonics sonochemistry 42 (2018) 411-421

For acoustic streaming please refer to: GSB Lebon “Ultrasonic liquid metal processing: The essential role of cavitation bubbles in controlling acoustic streaming” Ultrasonics sonochemistry 55 (2019) 243-255

“....of conventional casting becomes more difficult, so the ultrasonic casting process is introduced in the casting process” please improve this sentence

“The maximum vibration amplitude measured on the end face of the ultrasonic generator is 20μm” is it peak to peak?

“Materials and Methods should be described with sufficient details to allow others to replicate and build on the published results. Please note that the publication of your manuscript implicates that you must make all materials, data, computer code, and protocols associated with the publication available to readers. Please disclose at the submission stage any restrictions on the availability of materials or information. New methods and protocols should be described in detail while well-established methods can be briefly described and appropriately cited.” I guess this paragraph shouldn’t be there!!! Please amend

What the black square indicates above the vacuum pump is figure 1?

“1000±50w[16], In the furnace cavity” After comma you normally start with a small letter! Too many careless mistakes

From where have you obtained the nominal composition of the 2195 alloy?

Figure 3 the scale stamps are not visible and the quality is poor. Please replace photos with better quality and higher resolution

Figure 4, is this size grain reduction sufficient to justify the use of ultrasound? Grains are still large! Authors should explain this outcome

“The main improvement mechanism of USMT is summarized, cavitation-enhanced heterogeneous nucleation; heterogeneous nucleation particles and molten elements are homogenized under the action of ultrasound [17].” Please add this reference that provide insights into the cavitation enhanced nucleation via wetting, fragmentation and dispersion: A Priyadarshi “On the governing fragmentation mechanism of primary intermetallics by induced cavitation” Ultrasonics Sonochemistry 70 (2021) 105260; F. Wang “In situ observation of ultrasonic cavitation-induced fragmentation of the primary crystals formed in Al alloys” Ultrasonics sonochemistry 39 (2017) 66-76; doi:10.1016/j.ultsonch.2015.04.029

“At the moment when the cavitation bubble collapses, a local high temperature and high-pressure pulse is generated inside the bubble” please amend this is not inside the bubble! You can find some values to refer to liquid micro jets and shock waves in these 2 papers: doi:10.1016/j.ultsonch.2013.10.003 and Ultrasonics sonochemistry 21 (2014) 866-878 and M. Khavari “ Characterization of shock waves in power ultrasound” Journal of Fluid Mechanics (2021) 915

Figure 5 precipitate phases and particles are not clear and needs to be shown within the actual photos!

“The SEM image shows that obvious pores and loose defects are found in the cross section of the ingot without USMT....” these characteristics needs to be indicated within the actual photos of figure 8...as it stands with lack of detail description photos in figure 8 looks very similar among both cases....and results are not clear.....

Author Response

Dear editors and reviewers,

Thank you a lot for your helpful comments and constructive suggestions on our manuscript. We have learned the comments and suggestions earnestly and made careful corrections which we hope to gain your approval.

The responses to the reviewer’s comments and the main corrections are listed point by point as follows, Please see the attachment.

Round 2

Reviewer 1 Report

Quenstions raised by the reviewer were clarified and the manuscript was thoroughly revised. (Only OM images with polarized light were not visible on the revised form of the manuscript.) I think that the paper can be now publishable.

Reviewer 2 Report

Authors have sufficiently answered all the comments and manuscript can now be accepted for publication. However manuscript needs some attention as many of the figures are not shown properly such as figure 3 or have not been properly aligned see figure 9. Maybe the Editorial office can fix that issue if this is related with the conversion of the word file to pdf.